# Solution Precursor Plasma Spraying of TiO_2_ Coatings Using a Catalyst-Free Precursor

**DOI:** 10.3390/ma16041515

**Published:** 2023-02-11

**Authors:** Key Simfroso, Shena Ramyr Cabo, Romnick Unabia, Angelito Britos, Paweł Sokołowski, Rolando Candidato

**Affiliations:** 1Department of Physics, College of Science and Mathematics, Mindanao State University-Iligan Institute of Technology (MSU-IIT), Andres Bonifacio Ave., Tibanga, Iligan City 9200, Philippines; 2Premier Research Institute of Science and Mathematics (PRISM), Mindanao State University-Iligan Institute of Technology (MSU-IIT), Andres Bonifacio Ave., Tibanga, Iligan City 9200, Philippines; 3Department of Metal Forming, Welding and Metrology, Faculty of Mechanical Engineering, Wrocław University of Science and Technology (WUST), Wybrzeże Stanisława Wyspiańskiego 27, 50-370 Wrocław, Poland

**Keywords:** titanium dioxide, solution precursor plasma spraying, microstructure

## Abstract

The microstructural characteristics and phase composition of solution precursor plasma-sprayed (SPPS) titania-based coatings using a catalyst-free precursor are reported in this work. An ethanol-based solution containing titanium isopropoxide was used to deposit TiO_2_ coatings. The thermal behavior of the solution precursor changed as its phase transformation temperature increased when the molar concentration was increased from 0.3 M to 0.6 M. Scanning electron micrographs showed that the surface of the coatings was composed of nano- and submicron-sized spherical particles (<1 μm) with sintered and melted particles. The cross-sections showed a porous structure using lower concentrations and dense coating formation with micropores using higher concentrations, with thicknesses of about 5 µm–8 µm. Moreover, the coatings when the number of spray passes was increased were 16 µm–20 µm thick, giving an average layer thickness of 0.6 µm deposited per spray pass in all cases. Phase analysis revealed the presence of both the anatase and rutile phases of TiO_2_ in coatings sprayed with various concentrations at various stand-off distances. More detailed discussion is presented with respect to the effects of the solution concentration, stand-off distance, and number of spray passes on the coating’s phase composition and microstructure.

## 1. Introduction

Recently, coatings with micro- and nano-scale structural characteristics have drawn a lot of interest in photocatalysis due to their impressive surface area, which serves to provide active sites and facilitates efficient photoabsorption. Accordingly, if the absorption occurs at the boundary of the crystal, an indirect electron transition—which can significantly increase the absorption of light—could take place [1]. This occurs when a material has nanocrystals with a high surface-to-volume ratio and a sufficient amount of surface atoms. In photocatalysis, this gives nanoparticles in the few-nanometer size regimes an added advantage.

Titanium dioxide (TiO_2_) has been regarded as one of the most promising metal oxide semiconductors (MOSs) due to its unique mechanical and chemical resistance and photocatalytic properties [2]. The photocatalytic activity of TiO_2_ coatings depends strongly on their phase composition and microstructure. Many techniques have been employed to tailor these properties of TiO_2_, which include sol–gel methods [3], spray pyrolysis [4], chemical vapor deposition (CVD) [5], cold spraying [6], high-velocity oxy-fuel (HVOF) [7], and plasma spraying processes [8,9,10]. Among these methods, the plasma spraying process is on the cutting edge of flexible and high-rate deposition, in which the thickness and morphology can be precisely controlled [10,11]. Specifically, solution precursor plasma spraying (SPPS) was introduced to obtain coatings with micro- and nano-scale structural features.

SPPS is a relatively novel technique that has been used to deposit metal oxide coatings such as ceria (CeO_2_) [12], alumina–zirconia (Al_2_O_3_–ZrO_2_) [13], tungsten oxide (WO_3_) [14] and zinc oxide (ZnO) [15,16]. The SPPS coatings typically possess a highly porous structure and a high specific surface area, and their thickness can be tailored, ranging from a few hundred nanometers to tens of microns. Chen et al. [17] successfully obtained a dense TiO_2_ coating purely composed of rutile phases via the SPPS process, using a highly concentrated solution. They also obtained a porous TiO_2_ coating in which the anatase and rutile phases of TiO_2_ coexisted by lowering the solution concentration and adding a catalyst precursor at the same time [18]. Supplemented with a catalyst, pure SPPS TiO_2_ coatings—in which the anatase and rutile phases coexist—were also successfully produced by Du et al. [10] and Adán et al. [19]. Aruna et al. [11] obtained SPPS TiO_2_ coatings with spherical splats and unmelted particles with the use of a catalyst, finding excellent photocatalytic activity. Moreover, phase transformation can be avoided by controlling the processing temperature of the plasma jet in SPPS. In fact, Chen et al. further observed how the plasma power can influence the phase content of TiO_2_ coatings [18]. 

Clearly, varying the operational spray parameters during plasma spraying of a TiO_2_ solution derived from a catalyst-free precursor has not yet been performed, and it is not known how this operation could influence the microstructural properties of the SPPS coatings. Thus, the approach of this work entails the comparison between the outcomes of varying operational parameters—such as the solution concentration, stand-off distance, and number of spray passes—when spraying TiO_2_ solution using a catalyst-free precursor. A typical issue in SPPS is corrosion of the spraying equipment, which can be brought on by a precursor catalyst. In order to minimize this issue, it is advisable to avoid using an acidic catalyst.

The current contribution is focused on the analysis of the microstructural characteristics and phase composition of plasma-sprayed TiO_2_ coatings obtained using a catalyst-free precursor. Knowing the features of the coatings will enable us to choose the optimal operational spray parameters and produce coatings with the desired photocatalytic qualities. A workable method for producing nanostructured TiO_2_ coatings with effective deposition and controlled anatase content using plasma spraying of a catalyst-free TiO_2_ solution precursor is also presented in this work.

## 2. Materials and Methods

The following methods were performed, broken down into three categories: solution preparation, plasma spraying process, and sample characterization. The first part lists the precursors for the preparation of catalyst-free TiO_2_ liquid feedstocks and presents the coating sample. The experimental setup for the solution precursor plasma spraying process is also described, along with the operational spray parameters utilized during the experiments. A schematic representation of the process employed to carry out the current work is shown in Figure 1.

### 2.1. Solution Preparation and Plasma Spraying

The solutions were prepared by vigorously stirring a volume of titanium isopropoxide (Ti(OCH(CH_3_)_2_)_4_) in an appropriate amount of ethanol for 30 min. This was a single-step process, with no addition of a catalyst to the solution. The concentrations of titanium isopropoxide in the resulting transparent precursor solutions were 0.3 M and 0.6 M. The samples obtained from these concentrations are hereafter abbreviated as FS1 and FS2, respectively. Another sample was FS3, which was a coating produced from a 0.6 M solution with an increased number of deposition passes. Table 1 enumerates the samples, indicating the specific parameter values used in their spraying.

TiO_2_ coatings were obtained by the use of solution precursor plasma spraying system using a Praxair SG-100 spray torch. Argon (Ar) was used as the primary gas, with a flow rate of 45 slpm, and the secondary gas used was hydrogen (H_2_), with a flow rate of 5 slpm. The liquid feedstock was fed to the nozzle from a closed tank. The liquid was radially injected as a continuous stream into the plasma jet 4 mm from the end of the jet through a 0.2 mm diameter orifice, which was 15 mm from the jet axis. The stand-off distances were varied at 40 mm, 50 mm, and 60 mm. A 28 kW plasma power was used to spray the samples. Single- and full-scan sprays of 10 and 30 passes were carried out, with a relative torch velocity of 500 mm/s. All coatings were deposited using the process conditions listed in Table 2. The coatings were deposited onto 3 mm thick, 25 mm diameter stainless steel disk substrates. The stainless steel substrates were preheated to a temperature of ~200 °C by exposure to the plasma jet before the liquid feedstock was applied. 

### 2.2. Characterization of Solution Precursor and Coatings

The solution samples were dried to harvest TiO_2_ powders [20] for differential thermal analysis–thermal gravimetric analysis (DTA–TGA) characterization to determine the thermal behavior of the materials. The samples were heated from room temperature to 1000 °C at rates of 5 °C/min and 10 °C/min in flowing nitrogen gas. The TiO_2_ coatings were mounted with epoxy resin and polished for microstructural observation and measurement of coating thickness via scanning electron microscopy (SEM). X-ray diffraction (XRD) was used to determine the composition of the crystalline phase of the coatings using Cu Kα radiation. The XRD spectra were acquired in the 10° to 70° (2θ) angle range.

## 3. Results

### 3.1. Thermal and Crystallization Behaviors of the As-Dried Solution Precursor

Using a DTA–TGA analyzer, the weight loss of the precursor as a function of temperature was measured after the precursor was dried at room temperature. The typical DTA–TGA curves for the crystallization of the TiO_2_ precursor obtained at heating rates of 10 °C/min in nitrogen are shown in Figure 2. The DTA curves of each sample show an endothermic peak at ~80–100 °C, with a corresponding mass change of 27.6% for the 0.6 M TiO_2_ and 19.0% for the 0.3 M TiO_2_, which was caused by the evaporation of the solvent present in the material. The endothermic peak at ~260 °C, with a mass change of 5.8% and 3.7% for 0.6 M and 0.3 M TiO_2_, respectively, can be attributed to the decomposition of organic materials. Crystallization of the amorphous TiO_2_ to anatase TiO_2_ can be attributed to the exothermic peak at temperatures between 400 °C and 425 °C, as verified in its first derivation. The crystallization temperatures were comparable to the temperatures reported elsewhere [17]. Moreover, there were no corresponding weight losses during crystallization, which was expected not to happen during this process. The formation of rutile, on the other hand, was clearly visible when the powders were heated at 5 C°/min, as shown in Figure 2b. The anatase–rutile transformation temperatures for 0.3 M and 0.6 M TiO_2_ were ~713 °C and ~812 °C, respectively.

### 3.2. Single-Pass Spray Deposits

The morphology of the deposits from a single-pass spraying is presented in Figure 3. Figure 3a,b show the deposits from single-pass spraying of the TiO_2_ solution feedstock using different stand-off distances. The micrographs show sintered nano- and submicron-sized particles and agglomerated submicron-sized particles. Additionally, spherical particles can be seen. No splats can be observed for any of the single-pass spray deposits at different stand-off distances. These deposits observed on the surface of the substrate are characteristic features of plasma-sprayed coatings using solution precursors. As presented in Figure 3c, the elemental composition of the deposits along the surface of the coating (arrow indicated in Figure 3a) displays the presence of 42.1% Ti and 46.3% O, confirming the potential of the solution plasma spraying process for depositing TiO_2_ coatings.

### 3.3. Full-Pass Spray Coatings

#### 3.3.1. Microstructure of Coatings

Figure 4 depicts TiO_2_ coatings after 10 passes of full spraying of 0.3 M solution precursor. The surface of the coating is made up of nano-/submicron-sized spherical particles (<1 μm) that have been sintered and melted. The surface images of the FS1 coatings in Figure 4a–c reveal uneven surfaces that match the cross-sections of FS1 in Figure 4d–f. The cross-section of the obtained coatings shows a microstructure with micropores and a thickness of about 5 µm–8 µm. It can be observed that when the stand-off distance increases, the coating thickness decreases. Figure 4g displays the FS1SD40 coating’s internal structure. Only a small portion of the cross-section is visible in this location, revealing dense and granular regions of the porous FS1SD40 coating that resemble a two-zone microstructure.

To further establish the appropriate spray parameters for solution precursor plasma spraying of TiO_2_ feedstocks, 0.6 M TiO_2_ solution was sprayed at different stand-off distances. The cross-sections of the FS2 coatings in Figure 5d–f show nearly even surfaces, which match their surface images with the uniform particle sizes shown in Figure 5a–c. Similarly, as in the case of 0.3 M TiO_2_, it can be observed in Figure 5 that as the stand-off distance increases, the thickness of the coating decreases. The presence of lighter spots within the coating can be observed, and this might be attributable to unprocessed particles. Comparing the cross-sectional images of the coatings sprayed using different solution concentrations, a coating with a dense microstructure was developed using the 0.6 M concentration. In contrast, a relatively porous microstructure was developed using the 0.3 M concentration. The internal structure of the dense coating is presented in Figure 5g. It also portrays grainy particles and dense areas, which can be compared to a two-zone microstructure.

Thicker coatings were obtained when the number of spray passes was increased, as shown in Figure 6. Dense coatings were also obtained using the same concentration as FS2. The average thicknesses of the FS3 coatings were 15.8 ± 0.47 µm, 19.2 ± 0.56 µm, and 20.3 ± 0.88 µm with increasing stand-off distance, indicating an average layer thickness of 0.6 µm deposited per spray pass. Figure 4 and Figure 5 show that the stand-off distance did not affect the coating microstructure but only the coating thickness. However, for the coatings in Figure 6, the coating microstructure was affected by varying the stand-off distance. A sponge-like region was formed that made the FS3SD60 coating thicker, as shown in Figure 6c. This appearance of this region was caused by the repeated heat treatment because of the increased number of spray passes.

#### 3.3.2. XRD Analysis

The analysis of the XRD diffractograms for all coatings was carefully carried out with the accompaniment of ICSD patterns of the anatase (red) and rutile (blue) phases of tetragonal TiO_2_. The XRD patterns of the FS1 and FS2 coatings with varied stand-off distances are shown in Figure 7. The coatings are polycrystalline, composed of anatase and rutile phases. This also demonstrates that increasing the solution concentration results in a decrease in the anatase (101) peak intensity and an increase in the rutile (110) peak intensity. Moreover, the austenite peak intensities for the stainless steel substrate at 2*θ* = 43° (111) and 2*θ* = 54° (200) decreased when the solution concentration was increased. On the other hand, the peak at 35° is typical of TiO_2_-B [21], which was detected in the XRD patterns of FS1, FS2, and FS3 coatings in Figure 7 and Figure 8. TiO_2_-B is a crystalline form of titania with a looser structure than that of anatase and rutile.

The rutile and anatase phase percentages of the coatings obtained at varying stand-off distances for different concentrations were calculated. The anatase content ($f_{A}$) in the as-sprayed TiO_2_ coatings was determined by the following equation [11,18,22]:(1)$$f_{A}=\frac{I_{A}}{I_{A}+1.26I_{R}}.$$
where $I_{R}$ and $I_{A}$ are the XRD peak intensities of rutile (110) at 2*θ* = 27.4° and anatase (101) at 2*θ* = 25.4°, respectively. Using a solution concentration of 0.3 M, there was 69.7% anatase and 30.8% rutile for the FS1SD40 coating, 58.5% anatase and 41.4% rutile for the FS1SD50 coating, and 44.1% anatase and 55.9% rutile in the as-sprayed FS1SD60 coating. With the increase in the solution concentration, the rutile content increased. The coatings sprayed using a solution concentration of 0.6 M were composed of 10.7% anatase and 89.3% rutile for FS2SD40, 10.8% anatase and 89.1% rutile for FS2SD50, and 14.9% anatase and 85.1% rutile for FS2SD60. It can be partially concluded that with the increase in the concentration of the solution feedstock, the rutile content increases and the anatase content decreases, as observed in Figure 9.

Moreover, the TiO_2_ coatings obtained by increasing the number of deposition passes portrayed less pronounced peak intensities for substrates, as shown in Figure 8, implying thicker coatings and consistent with the SEM images in Figure 6. It can also be seen that the anatase (101) peak increased and the rutile (101) peak decreased as the stand-off distance increased. The anatase phase percentages were 11.3%, 21.6%, and 26.4% with increasing stand-off distance. Consequently, the rutile phase percentages decreased with increasing stand-off distance: 88.7%, 78.4%, and 73.6%, respectively. From Figure 9, it can be inferred that with the higher concentration, when the number of spray passes increases, the rutile content decreases and the anatase content increases. With more spray passes, the ratios between rutile and anatase did alter, but at higher solution concentrations and all stand-off distances the rutile phase content was always greater than the anatase phase content.

Based on the measured width of the diffraction curves of the crystal planes, the crystallite sizes were estimated using the Scherrer formula [23]:(2)$$t_{\left(hkl\right)}=\frac{0.9\lambda }{\beta \cos\theta_{B}}.$$
where $t_{\left(hkl\right)}$ is the crystallite size; λ is the wavelength of X-ray radiation, which is equal to 0.15045 nm; β is the full width at half-maximum (FWHM) of the peak intensity; and $\theta_{B}$ is the Bragg angle of reflection of a specific crystal plane (hkl). Figure 10 presents the calculated crystallite sizes, and they are all less than 30 nm. For lower concentration, the crystallite sizes of anatase are always larger than those of rutile; however, there is a wide distribution of the crystal size values. On the other hand, the rutile crystallite sizes are always larger than those of anatase at the higher concentration, and the crystal size is homogenous at lower stand-off distances. Figure 10 also shows that changing the solution concentration from 0.3 M to 0.6 M decreased the crystallite sizes of anatase and increased the crystallite sizes of rutile for the same stand-off distance. Conversely, there was no pattern to the crystallite sizes for any of the FS3 coatings.

## 4. Discussion

The photocatalytic activity of TiO_2_ is primarily dependent on its microstructure and phase composition. Thus, it is important to control the phase composition of the TiO_2_ coating. This could be accomplished by controlling the plasma heat, since the precursor droplets are assumed to undergo physical and chemical changes during the SPPS process within a brief in-flight duration in a high-temperature plasma jet. Hence, the transformation temperature of the material should be monitored.

As indicated in the DTA–TGA analyses in Figure 2, the solution precursor undergoes solvent vaporization, decomposition, crystallization, and phase transition. The anatase formation temperature was 400 °C and 425 °C for powders derived from 0.3 M and 0.6 M concentration solution, respectively. Pyrolysis happened below 350 °C for both precursors, and both crystallized at ~400 °C, which is similar to the temperatures in previous reports [18,24]. The exothermic peaks centered at ~713 °C and ~812 °C, associated with the transition from anatase to rutile, are temperatures comparable to the reported transition temperature [17]. However, the rutile transition temperature was higher for the higher-concentration solution than the lower-concentration solution. During thermal analysis, the temperature at which the maximum deflection is observed varies with the heating rate for certain types of reactions. The molar concentration of the reactant solution played an important role in the crystallite formation process. Because higher concentrations correspond to higher saturation, this results in greater driving forces for crystallization [25]. When DTA–TGA is used, more energy is required for both complete crystallization and full phase change for powders made from more concentrated solutions, leading to higher transformation temperatures being detected. Based on the DTA results, it is anticipated that anatase and rutile phases will coexist in the coatings when utilizing 28 kW of plasma power [18], because anatase begins to transform into rutile at temperatures higher than 600 °C. Furthermore, the rutile crystal phase is more likely to dominate when a higher solution concentration is utilized [17]. These hypotheses were realized and verified by the XRD analyses shown in Figure 7, Figure 8 and Figure 9.

Figure 4 shows a porous type of coating. These coating features, where rougher surfaces and smaller particles are observed in the absence of a precursor catalyst, are similar to those described in [11]. This can be supported by the X-ray diffractogram of FS1, which shows sharper and stronger peaks for the substrate, indicating a penetrable, porous coating. Regardless of the precursor, whether a catalyst is present or not, a porous coating may be developed—especially for lower solution concentrations. For these coatings, when the stand-off distance increases, the coating thickness decreases. Some of the solution droplets inside the plasma jet might not have enough momentum to be deposited onto the substrate, so for longer spray distances only a few melted particles were propelled to the substrate and formed the coating.

The coatings obtained at the higher concentration showed a dense, compact structure with micropores, as further supported by the XRD patterns of FS2, which depict weaker substrate peaks. The solution particles possibly went through early saturation and started to precipitate at an earlier time. Early precipitation may occur because of the increased droplet temperature caused by the increased heating of plasma, influenced by the vaporization of a solution with a higher concentration [26,27]. Fine-grained particles of less than 1 μm in size were apparent when looking at the internal structure of the coating. The size of the particles largely depends on the evaporation outcomes. The solvent used was ethanol, which experiences rapid vaporization, resulting in a fast decrease in particle size [28]. It was noted that a grainy coating structure was produced instead of the columnar structure commonly formed by layers of molten splats. The selected concentrations were much lower compared to those reported by Chen et al. [17]. Volumetric precipitation is favorable when the solution concentration is higher [29,30]. Hence, molten particles are likely to deposit on the coating surface [18], accompanied by constant heat treatment of the plasma jet, and layers of splats are more likely to occur. The dense, fine-grained coating structure occurred upon the deposition of semi-molten particles.

Additionally, utilizing the same number of spray passes, FS2 was thinner than the other deposits. This was predicted through the DTA results, in which the crystallization temperature increased when the higher concentration was utilized. According to reports, the crystallization temperature is strongly dependent on the coating thickness [31,32,33]—the thinner the coating, the higher the crystallization temperature of TiO_2_.

While it was foreseen that the coating thickness would increase with an increased number of spray passes, as portrayed by the FS3 coatings, the coating thickness also increased with increasing stand-off distance because of the increased porosity. The semi-molten particles are likely to resolidify while in flight, especially at greater stand-off distances. The increased number of spray passes also implies repeated scanning of the coating and longer exposure to the plasma jet. This may cause in situ evaporation of the resolidified particles upon impact, resulting in the formation of the sponge-like structure visible in Figure 6c, making the coating thicker than at lower stand-off distances.

Since the FS2 samples depicted more pronounced rutile peaks than FS1, this means that a higher fraction of rutile TiO_2_ was present in the sprayed coating when a higher concentration was utilized. These results are supported by the DTA results of the powders when using a 0.6 M solution concentration, which showed that the transformation of amorphous TiO_2_ into anatase TiO_2_ occurred at a higher temperature compared to that with the 0.3 M solution. No other parameters were changed, such as plasma power, stand-off distance, or primary and secondary gas flow rates. However, according to Chen et al. [29], the concentration of the precursor caused slight changes in solution specific mass, surface tension, and a broad change in solution viscosity. These changes do not affect the average size of the droplets, but only the behavior of a concentrated solution droplet in a plasma environment. The solution precursor is a mixture of TIP and ethanol that will undergo alcoholysis and then decompose in the presence of plasma, as expressed in the following equations:(3)$$\mathrm{Ti}{\left({\mathrm{OC}}_{3}H_{7}\right)}_{4}+4{\mathrm{CH}}_{3}{\mathrm{CH}}_{2}\mathrm{OH}\to \mathrm{Ti}{\left({\mathrm{OCH}}_{2}{\mathrm{CH}}_{3}\right)}_{4}+4C_{3}H_{8}O$$
(4)$$\mathrm{Ti}{\left({\mathrm{OCH}}_{2}{\mathrm{CH}}_{3}\right)}_{4}\overset{\mathrm{heat}}{\to }{\mathrm{TiO}}_{2}+\mathrm{hydrocarbons}$$

The effect of solvent vapors on the hot gas transport properties must also be taken into account [34]. Because of the high concentration of the TiO_2_ solution and the high temperature of the plasma, rapid vaporization of alcohol in the solution droplet takes place, which causes an increase in the ability of heating factor (AHF) of the plasma jet due to an increased number of species participating in the collision interactions between plasma species [27,35,36]. The higher the AHF of the plasma, the faster it can reach a high temperature, which may lead to a higher temperature transition of amorphous TiO_2_ into crystalline TiO_2_. Hence, the critical value of solute saturation may be achieved immediately after feeding the liquid into the jet and causing the early particle solidification of a solution droplet. The quicker the solid particle formation, the quicker the formation of crystalline TiO_2_ upon deposition [37]. It is then presumed that the rapid phase transformation from anatase to rutile will subsequently take place. Thus, the decrease in anatase crystallite sizes and the increase in rutile crystallite sizes were observed from the calculations based on XRD results with increasing solution concentration. The high-temperature plasma jet can thus be assumed to be useful in obtaining a highly crystalline coating.

Increasing the stand-off distance implies a longer dwell time of the particles in the plasma jet. Moreover, the increase in temperature is faster with a shorter stand-off distance. Beyond the temperature of 600 °C, nanocrystalline anatase starts to transform into rutile TiO_2_ [38]. Anatase is more likely to change entirely into rutile as a result of the plasma treatment and the high surface energy of anatase [10,38,39], especially when the coating is at a closer stand-off distance. The coating steadily experiences more heat when it is closer to the plasma jet.

This method of plasma spraying for a catalyst-free TiO_2_ solution precursor appears to be viable for producing nanostructured TiO_2_ coatings with excellent deposition and controlled anatase content.

## 5. Conclusions

TiO_2_ coatings were deposited from an ethanol-based, catalyst-free solution containing titanium isopropoxide using the SPPS process. The use of catalyst-free precursors can provide dense coatings made of fine-grained particles at the nano-scale, which may be advantageous for photocatalytic applications. Varying a few of the operational spray parameters resulted in different coating properties. The solution precursor concentration had a significant effect on the sprayed coating structure and phase composition. Compared to coatings sprayed using a lower concentration, coatings with a higher solution concentration showed the following characteristics: decreased thickness, decreased anatase content and crystallite sizes, and increased rutile content and crystallite sizes. A thicker coating and more anatase content were produced by increasing the stand-off distance when the number of spray passes was increased. By combining the process parameters, our results suggest that the properties of TiO_2_ coatings can be manipulated for photocatalytic application. Moreover, these initial results may be of great interest to researchers seeking to engineer SPPS TiO_2_ coatings with controlled structure and composition by adding a foreign ion to a catalyst-free TiO_2_ solution precursor.

## Figures and Tables

**Figure 1 materials-16-01515-f001:**
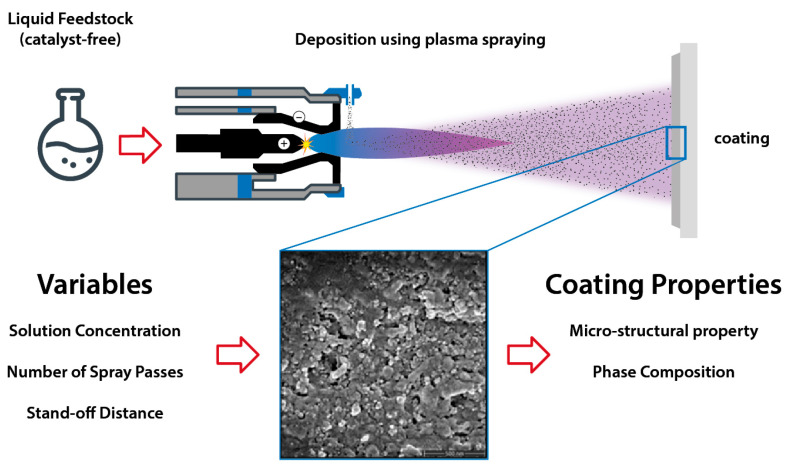
Process flowchart demonstrating the SPPS deposition of TiO_2_ solution using a catalyst-free precursor with varied spray parameters and their influence on specific coating properties.

**Figure 2 materials-16-01515-f002:**
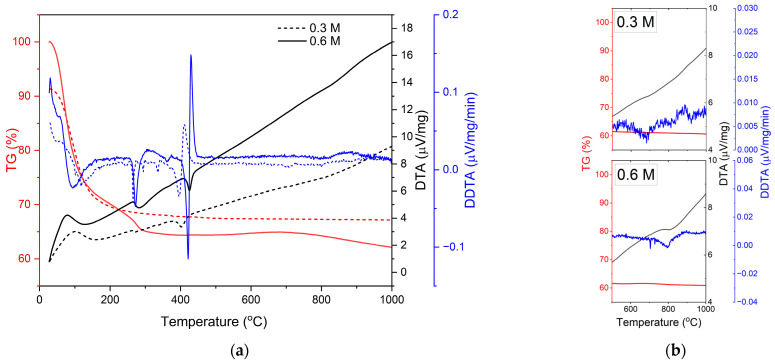
Typical DTA–TGA curves of dried TiO_2_ precursor powders at a heating rate of (**a**) 10 °C/min and (**b**) 5 °C/min. Graphs of the first derivation of the DTA results are included.

**Figure 3 materials-16-01515-f003:**
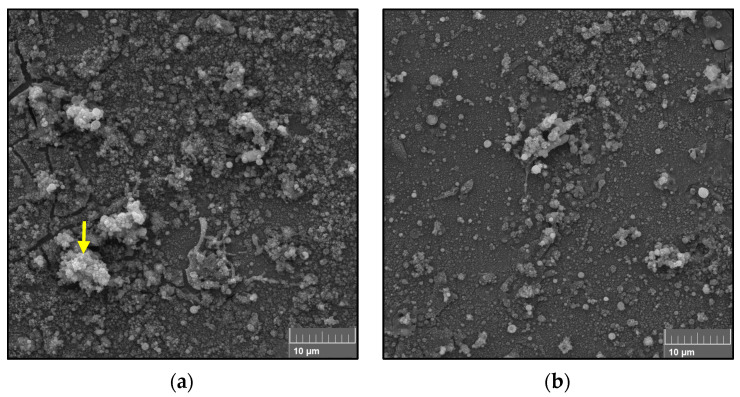
SEM micrographs of single-pass sprayed (0.3 M) TiO_2_ solution using different stand-off distances: (**a**) 40 mm; (**b**) 60 mm. (**c**) Elemental composition of single-pass spray deposits at a 40 mm distance (arrow indicated in (**a**)).

**Figure 4 materials-16-01515-f004:**
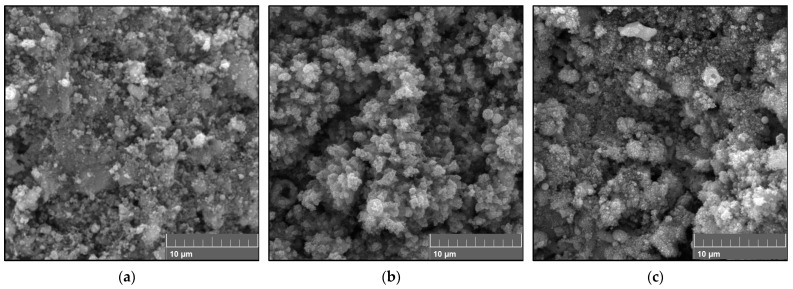
SEM micrographs of the surface of coatings obtained from full-pass sprayed (0.3 M) TiO_2_ solution and their cross-sections at (**a**,**d**) 40 mm, (**b**,**e**) 50 mm, and (**c**,**f**) 60 mm stand-off distances. (**g**) The cross-sectional coating structure of FS1SD40 at higher magnification.

**Figure 5 materials-16-01515-f005:**
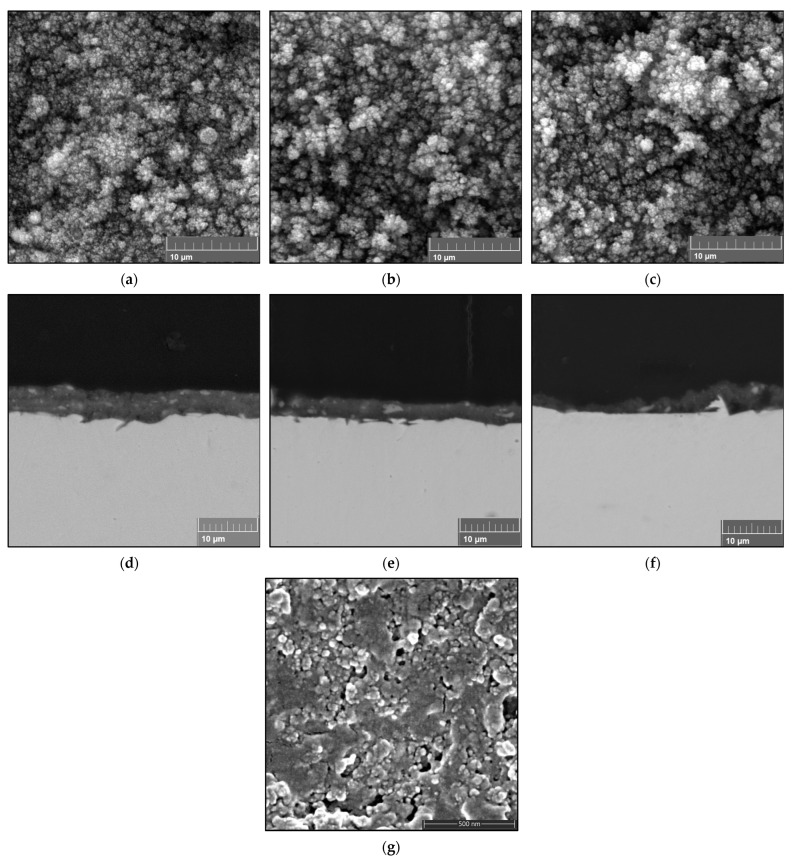
SEM micrographs of the surface of coatings obtained from full-pass sprayed (0.6 M) TiO_2_ solution and their cross-sections at (**a**,**d**) 40 mm, (**b**,**e**) 50 mm, and (**c**,**f**) 60 mm stand-off distances. (**g**) The cross-sectional coating structure of FS2SD40.

**Figure 6 materials-16-01515-f006:**
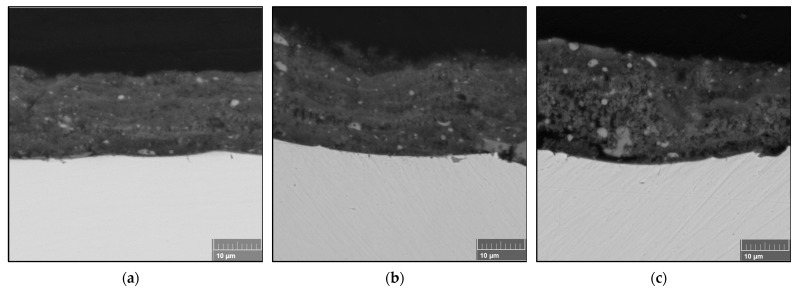
Cross-section of the coatings obtained from full-pass sprayed (0.6 M) TiO_2_ solution when the number of passes was increased to 30 at (**a**) 40 mm, (**b**) 50 mm, and (**c**) 60 mm stand-off distances

**Figure 7 materials-16-01515-f007:**
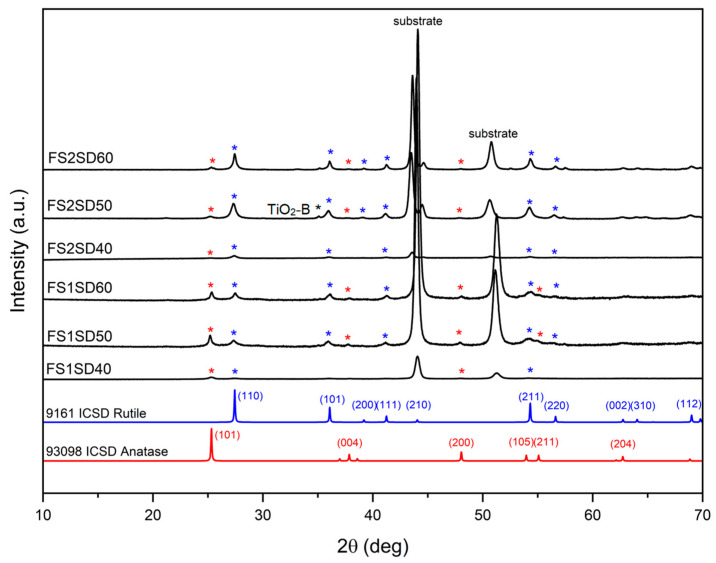
XRD of SPPS TiO_2_ coatings derived from full-pass sprayed (0.3 M) TiO_2_ solution at different stand-off distances. (The red, blue and black asterisks (*) correspond to the signature peaks of anatase TiO_2_, rutile TiO_2_ and TiO_2_-B, respectively, which are visible in each pattern when plotted individually).

**Figure 8 materials-16-01515-f008:**
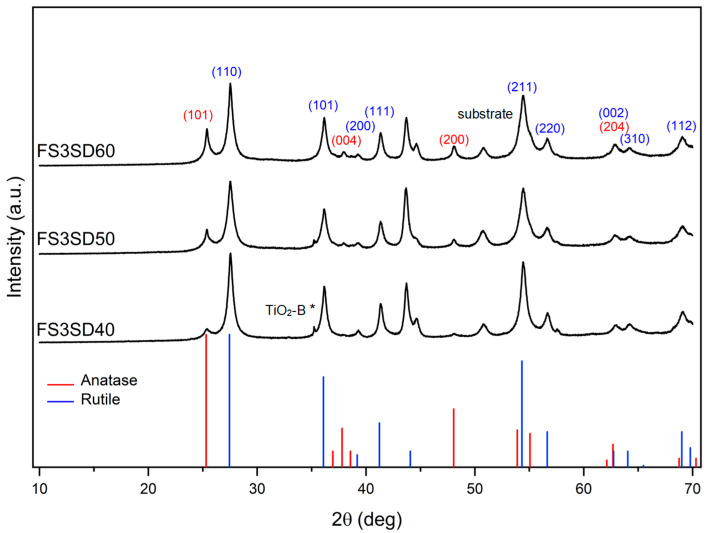
XRD patterns of full-pass sprayed TiO_2_ coatings using 0.6 M precursor concentration and increasing numbers of spray passes. (The black asterisk (*) corresponds to TiO_2_-B.)

**Figure 9 materials-16-01515-f009:**
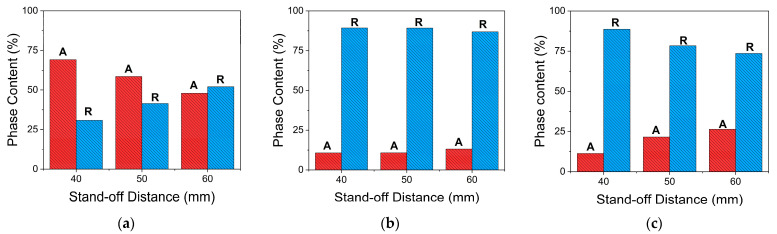
Anatase (A) and rutile (R) phase content percentages in (**a**) FS1, (**b**) FS2, and (**c**) FS3 coatings as a function of stand-off distance.

**Figure 10 materials-16-01515-f010:**
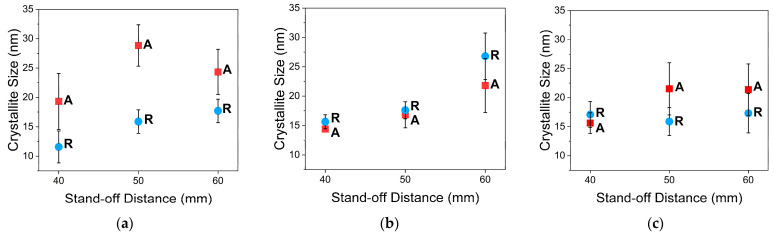
Calculated anatase (A) and rutile (R) crystallite sizes for (**a**) FS1, (**b**) FS2, and (**c**) FS3 coatings.

**Table 1 materials-16-01515-t001:** Sample determination according to varying solution concentrations and stand-off distances.

Sample	Concentration	No. of Passes	Stand-Off Distance	Sample Label
	0.3 M	10	40 mm	FS1SD40
FS1			50 mm	FS1SD50
			60 mm	FS1SD60
	0.6 M	10	40 mm	FS2SD40
FS2			50 mm	FS2SD50
			60 mm	FS2SD60
	0.6 M	30	40 mm	FS3SD40
FS3			50 mm	FS3SD50
			60 mm	FS3SD60

**Table 2 materials-16-01515-t002:** Solution precursor plasma spraying parameters for spraying TiO_2_ coatings.

Parameters	Value
Plasma power	28 kW
Ar flow rate	45 slpm
H_2_ flow rate	5 slpm
Torch velocity	500 mm/s
Feedstock feed rate	~35 g/min
Stand-off distance	40, 50, 60 mm
Substrate	Stainless steel
Number of spray passes	10, 30
Nozzle diameter	0.2 mm

## Data Availability

Not applicable.

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
