# Peer review of "Solution Precursor Plasma Spraying of TiO2 Coatings Using a Catalyst-Free Precursor"

_materials, 2023, doi:10.3390/ma16041515_

Round 1
Reviewer 1 Report
This manuscript reported that the microstructure and phase composition of solution precursor plasma plasma spraying of TiO2 coatings using catalyst- free precursor. The comments are following as:
1. In Figure 1, typical DTA-TG curves of dried TiO2 precursor powders at a heating rate of 10 ℃/min and 5 ℃/min were shown. However, the measured temperature range at a heating rage of 5 ℃/min is 550-1000 ℃, which is different at a heating rate of 10 ℃. It is difficult to obtain the phase transition temperature at about 260 ℃. In addition, these curves need to be clearly signed in Figure 1(a).
2. In Figure 6 and Figure 7, the XRD patterns of TiO2 coatings were shown, however, some diffraction peaks (e.g. 35Ëš for FS2SD50 in Figure 6 and 52 Ëš in Figure 7) are not analyzed, which are not belonged to those of anatase and rutile phases.
3. In Figure 8 and Figure 9, some symbols are not clear, for example, ‘A’ means anatase phase, and ‘R’ means rutile phase?
4. The text shows many grammatical, syntax or word usage errors, which should be correct them carefully.
Reviewer 2 Report
In this manuscript, the authors presented a study on solution precursor plasma spraying of TiO2 coatings. They applied DTA-TGA, SEM, and XRD to characterize samples, and found that solution precursor concentration has a significant effect on the sprayed coating structure and phase composition.
In general, this research is pretty rough and lacks some critical pieces of evidence. The results could be presented better. This work may be reconsidered after major revision. I have included some comments hereby:
1. As to the DTA-TG characterization, the authors took the measurements after the precursor was dried at room temperature, while the concentrations of two precursor solutions were 0.3 M and 0.6 M. If they are dried with only solutes remaining, how can the thermal behaviors of the as-dried solution precursors be so different? Both are supposed to be the same.
2. How can the authors precisely measure the thickness of these coating layers? All cross-section SEM images were taken under lean angles, not from a perpendicular view.
3. Which sample did the authors measure in figure 4d?
4. The experiment design is not that reasonable as the authors introduced three variables (concentration, number of passes, and stand-off distance). The scenario of 0.3 M samples with 30 passes was not included.
5. Line 159-162: “Comparing the cross-sectional images of the coatings sprayed using different solution concentrations, coating with dense microstructure was developed using 0.6 M concentration. In contrast, a relatively porous microstructure was developed using 0.3 M concentration.” The SEM images of 0.3 M samples in Figure 3(a-c) obviously have different scales with the one of 0.6 M in Figure 4(d). If the image scales are so significantly different, how can the readers be convinced that the conclusions made in the manuscript sound reasonable? Also, for a complete comparison, not just the cross-section areas, all corresponding top surface micrographs should be provided and discussed in the analysis as the authors varied processing parameters in each sample.
Reviewer 3 Report
The article is very well written. However, the following comments need to address:
1. Equations numbering is wrong. Equation (2) is used twice.
2. Highlight the contributions of work in the introduction section.
3. What further controls should be considered? Add a flowchart of methodology for easy understanding of readers.
4. The conclusions are consistent and described clearly. However, the future part of the work still needs to be presented. Add future directions in conlclusion.
5. Increase the resolutions of figures 2 to 5.
Round 2
Reviewer 1 Report
There are no any questions for this manuscript. I suggest it to be accepted now.
Author Response
Dear Reviewer 1,
On behalf of my co-authors, thank you for the opportunity to revise our paper. I believe that our paper was improved thoroughly by your helpful comments. I appreciate your acceptance of our paper for publication in the Materials journal.
Respectfully yours,
Key Simfroso
Reviewer 2 Report
The authors have answered all technical questions raised by this referee and made corresponding changes in the manuscript. The revised paper looks in good shape and is well organized. I think this work can be accepted.
Author Response
Dear Reviewer 2,
On behalf of my co-authors, thank you for allowing us to submit a revised draft of our manuscript. We appreciate the time and effort that you have dedicated to providing your valuable feedback on our paper. I believe that our paper was improved thoroughly by your helpful comments. I appreciate your suggestion for our paper to be accepted for publication in the Materials journal.
Respectfully yours,
Key Simfroso